

# Heavy metal concentrations in soil and ecological risk assessment in the vicinity of Tianzhu Industrial Park, Qinghai-Tibet Plateau

Juan Qi, Xin Lu, Ninggang Sai, Yanjun Liu and Wangyi Du

Key Laboratory of Grassland Ecosystem of Ministry of Education, College of Grassland Science, Gansu Agricultural University, Lanzhou, Gansu Province, China

## ABSTRACT

Industrial parks in China are centers of intensive chemical manufacturing and other industrial activities, often concentrated in relatively small areas. This concentration increases the risk of soil pollution both within the parks and in surrounding areas. The soils of the Tibetan Plateau, known for their high sensitivity to environmental changes, are particularly vulnerable to human activity. In this study, we examined the concentrations (mg/kg) of 10 metal elements (As, Cd, Cr, Cu, Hg, Mn, Ni, Pb, Se, and Zn) in soils at depths of 0–10 cm, 10–20 cm, and 20–30 cm from the surface at three distances (500 m, 1000 m, and 1500 m from the park boundary) on the east, south, west, and north sides of the Tianzhu Industrial Park on the Qinghai-Tibet Plateau. The concentrations of As, Cr, Mn, and Pb were close to the standard reference values for the Qinghai-Tibet Plateau, while Cu, Ni, Se, and Zn levels were found to be 1.6-2.2 times higher than the reference values. Cd and Hg concentrations were particularly concerning, at 8.0 and 6.5 times higher than reference values, respectively. The potential ecological risk indexes indicated persistent risk levels for Cd and Se across various directions and distances. Variations in soil depth and direction were observed for the concentrations of As, Cd, Hg, Pb, Se, and Zn, underscoring the need for regular or long-term monitoring. Cd, in particular, presents a significant hazard due to its high concentration and its propensity for uptake by plants in the study area.

## INTRODUCTION

Earth's soils are facing severe pollution challenges due to rapid urbanization and industrialization (*Zhou et al., 2022*). Among the pollutants, the accumulation of heavy metals in soils has emerged as a critical concern for both public health and ecosystem safety (*Xiao, Zong & Lu, 2015*; *Modabberi et al., 2018*; *Yang et al., 2022*). Industrial activities are a major contributor to heavy metal contamination, posing significant risks to the long-term sustainable use of land and environmental health (*Bonanno & Cirelli, 2017*; *Maiti & Rana, 2017*). This issue has become an increasing global concern (*Tepanosyan et al., 2017*).

Corresponding author
Juan Qi, qijuan0622@163.com

Industrial parks in China are designated economic zones that offer infrastructure, services, and incentives to businesses, facilitating manufacturing and other industrial activities (*Zhang et al., 2022a*; *Zhang et al., 2022b*; *Zhang et al., 2022c*). The high density of these parks in relatively small areas leads to a heightened risk of severe industrial pollution within the parks and their surrounding regions (*Bang et al., 2022*). Industrial discharges, such as exhaust gases, wastewater, and residues, may contain various heavy metals (*Han & Xu, 2022*; *Fang et al., 2023*). Surface soils near industrial enterprises serve as the primary sinks for these pollutants, including heavy metals that are hazardous and tend to accumulate in both the environment and living organisms (*Yang et al., 2022*; *Wu et al., 2022*). These metals readily diffuse or migrate through the environment and are notoriously difficult to remove. Some heavy metal complexes are water-soluble and can easily enter animal and human food chains, posing significant health risks (*Wang et al., 2019*). Consequently, understanding the distribution and environmental impact of heavy metals in the soils of industrial parks is of critical environmental importance.

The Qinghai-Tibet Plateau, often referred to as the "Water Tower of Asia" serves as a vital ecological barrier for China. However, the region is highly vulnerable and sensitive to human activities. In recent years, intensive industrial development on the Plateau and in surrounding areas has led to increased heavy metal contamination of its soil and water (*Li et al., 2023*). Key sources of this pollution include vehicle emissions (*Cai et al., 2023*), the use of fertilizers and pesticides (*Sun et al., 2023*), mining activities (*Rouhani, Skousen & Tack, 2023*), and the deposition of exogenous pollutants, such as atmospheric aerosols transported from outside the Plateau (*Kang et al., 2019*; *Lin et al., 2023*). For example, cadmium (Cd) and arsenic (As) concentrations in the water of the Bailong and Yellow Rivers, along with their tributaries on the Plateau, exceed standard reference values (*Du et al., 2021*). The elevated As levels are primarily linked to the widespread distribution of As-rich shales, while Cd contamination is likely related to industrial, agricultural, and transportation activities in the region (*Zhang et al., 2015*; *Yang et al., 2020*). Therefore, assessing the heavy metal content in soils and identifying their sources from industrial parks on the Tibetan Plateau are essential steps in evaluating metal pollution and its impact on the local environment and communities. Tianzhu Industrial Park is situated in the eastern part of the Qilian Mountains on the Qinghai-Tibet Plateau. The Qilian Mountains act as a natural barrier between the Qinghai-Tibet Plateau and the Hexi Corridor, characterized by its unique geographical features, diverse ecological environment, and rich mineral resources. Given its ecological significance, environmental protection within the industrial park is of paramount importance.

This paper presents a case study from the Tianzhu Industrial Park, which has been under development since 2009 and located in the eastern Qilian Mountains of the Qinghai-Tibet Plateau. The park hosts several chemical processing manufacturers, raising concerns about heavy metal accumulation in the surrounding soils and the potential eco-toxicological effects. The main objectives of this study are to determine the concentrations of arsenic (As), cadmium (Cd), chromium (Cr), copper (Cu), mercury (Hg), manganese (Mn), lead (Pb), selenium (Se), nickel (Ni), and zinc (Zn) in relation to industrial activities in the eastern Qilian Mountains. Additionally, the study aims to comprehensively assess

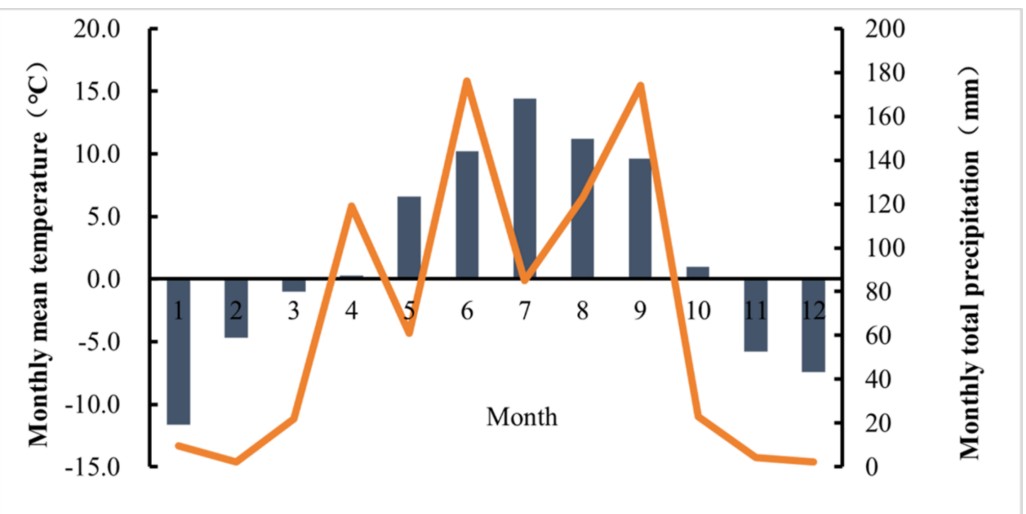

**Figure 1** **Monthly mean temperature and cumulative rainfall in 2021 at Tianzhu County, Gansu Province.**

the degree of metal contamination in the soils using established hazard indices, identify potential sources of heavy metals, and evaluate the ecological risks associated with toxic metal accumulation. The results are intended to provide valuable insights for developing soil pollution prevention and control strategies in industrial areas.

## MATERIALS & METHODS

### Study area

The study area is located near the Tianzhu Industrial Park (N 36°31′–37°55′, E 102°07′–103°46′) in the eastern Qilian Mountains, on the northeastern edge of the Qinghai-Tibet Plateau. The elevation of the area ranges from 2,040 m to 4,874 m. The distribution of monthly total precipitation and mean temperature is shown in Fig. 1. Mean monthly temperatures range from −11.6 °C in January to 14.4 °C in July, with an average annual temperature of 1.6 °C. Monthly precipitation varies from 2.1 mm in December to 176 mm in June, with most rainfall occurring during the plant growing season (June to September). There is no absolute frost-free period. The primary vegetation consists of alpine species, which are ecologically fragile and highly susceptible to environmental stress. The region experiences a typical mountain plateau climate, characterized by intense sunshine and significant seasonal variations in rainfall and humidity. The predominant soil type is mountain chestnut soil, with a pH range of 7.0 to 8.2.

### Sampling sites

In August 2021, soil samples were collected from three depth intervals (0–10 cm, 10–20 cm, and 20–30 cm) at 12 sites distributed across four cardinal directions (east, south, west, and north). These sites were located at distances of 500 m, 1000 m, and 1500 m from the industrial park, as shown in Fig. 2. The 500 m mark was defined as the boundary of

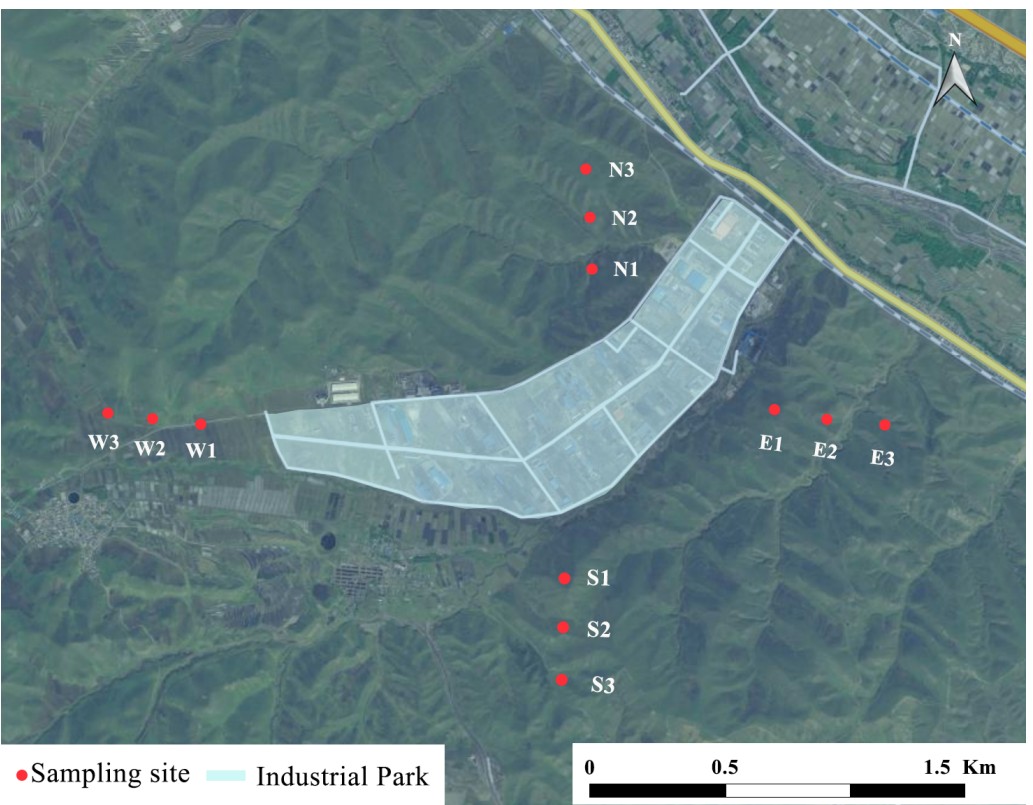

**Figure 2** **The study area and sampling sites.** These sites locate at distances of 500 m, 1000 m, and 1500 m from the four directions of the industrial park-East (E1, E2, E3), West (W1, W2, W3), South (S1, S2, S3), and North (N1, N2, N3). Maps were created in ArcGIS10.2 (Esri Inc., Redlands, CA, USA).

the last manufacturer in each direction. All sampling locations were accurately identified and recorded using a Trimble Geo XH 6000 handheld GPS, produced by Beijing Mingtu Technology Co., Ltd., Beijing, China.

## Sampling

At each sampling site, three replicate sampling points were selected, spaced approximately 20 m apart, allowing for triplicate samples to be collected at each location. Using a stainless-steel shovel, approximately 1 kg of soil from each depth interval was collected at each sampling point. After collection, the soil samples were thoroughly mixed and packed in zipper-sealed plastic bags. The samples were then transported to the laboratory and air-dried at room temperature for two weeks. Once dried, any visible gravel and plant roots were removed, and the soil was ground and passed through a one mm sieve for elemental content analysis.

## Chemical analysis

Precisely 0.5 g of soil was weighed and transferred into a 100 ml pressure-resistant polytetrafluoroethylene (PTFE) vessel for acid digestion using a TOP Wave Microwave Digester (Jena, Germany). The digestion process involved a mixture of $HNO_3$, HF, and

HClO$_4$ (all HPLC grade) in a 3:1:1 ratio at 120 °C until the solution became clear. Metal analysis was conducted in duplicate for each clarified digested sample. The concentrations of As, Se, Hg, Cd, and Pb were determined using an atomic fluorescence photometer (PF51, Beijing), while Cu, Cr, Mn, Ni, and Zn concentrations were measured using a flame atomic absorption spectrometer (ATS-990) (*Kiazai et al., 2019*; *Lu, 2021*).

## Evaluation of heavy metal pollution in soil

In this study, soil quality was evaluated using background values for metal elements from the Qinghai-Tibet Plateau (*Cheng & Tian, 1993*). The ecological risk assessment was performed using the Hakanson potential ecological hazard index (*Hakanson, 1980*), a widely used method to assess the degree of heavy metal pollution in soil (*Lv et al., 2014*). The index is calculated as follows:

$$C_f^i = C_s^i / C_n^i$$
$$E_r^i = T_r^i \times C_f^i.$$

where $C_f^i$ represents the contamination factor of individual heavy metals, $C_s^i$ is the concentration of metal i in the soil, and $C_n^i$ is the reference concentration of metal i. The potential ecological hazard coefficient for each heavy metal, $E_r^i$, is calculated using the toxic response factor $T_r^i$, which is set at specific values for different metals: 10 for As, 30 for Cd, 2 for Cr, 5 for Cu, Pb, and Ni, 15 for Se, 40 for Hg, and 1 for Zn and Mn (*Hakanson, 1980*; *Han & Xu, 2022*).

The potential ecological risk factor ($E_r^i$) is classified into 5 levels: low potential ecological risk ($E_r^i < 40$), moderate potential ecological risk ($40 \leq E_r^i \leq 80$), considerable potential ecological risk ($80 \leq E_r^i \leq 160$), high potential ecological risk ($160 \leq E_r^i \leq 320$), and very high ecological risk ($E_r^i > 320$) (*MacDonald, Ingersoll & Berger, 2000*; *Keshavarzi & Kumar, 2019*).

## Data processing and analysis

This study employed a 3 (soil depth) × 3 (distance away from the park boundary) × 4 (direction) factorial design. A general ANOVA model, including all interactions, was used to evaluate the effects of these factors on metal concentrations in the soil. Tukey's Honest Significant Difference test was applied to identify statistically significant differences between means, with $P < 0.05$ considered as the threshold for significance. Data were analyzed using SPSS Statistics 22.0 (IBM Corp., Armonk, NY, USA). The analysis indicated significant interactions between metal concentrations and the combined factors of soil depth, distance, and direction ($P < 0.05$). Further examination revealed that the variation among distances was relatively small compared to the variation due to soil depth and direction. Therefore, metal concentration data for the different distances were presented in a table, while the effects of soil depth and direction were illustrated in graphs created using ArcGIS 10.6 (Esri Inc., Redlands, CA, USA). Pearson correlation analysis was also performed to assess the relationships between metal concentrations.

**Table 1** Heavy metal concentrations (mg/kg) in soil.

|  | As | Cd | Cr | Cu | Hg | Mn | Ni | Pb | Se | Zn |
|---|---|---|---|---|---|---|---|---|---|---|
| Maximum | 56.4 | 0.92 | 119.2 | 46.4 | 0.23 | 831 | 88.6 | 53.4 | 0.88 | 162 |
| Minimum | 9.8 | 0.38 | 44.5 | 26.0 | 0.04 | 593 | 51.9 | 18.8 | 0.24 | 96 |
| Mean | 20.1 | 0.64 | 84.3 | 34.8 | 0.13 | 711 | 75.4 | 27.8 | 0.47 | 125 |
| Standard deviation | 12.2 | 0.15 | 18.5 | 4.9 | 0.06 | 67 | 7.8 | 8.1 | 0.13 | 15.2 |
| Reference values of heavy metals in soil in Tibet | 18.7 | 0.08 | 77.4 | 21.9 | 0.02 | 608 | 32.1 | 28.9 | NA | 73.7 |

**Table 2** Heavy metal concentrations (mg/kg) in soil at three distances from the boundary of Tianzhu Industrial Park.

| Distance, m | As | Cd | Cr | Cu | Hg | Mn | Ni | Pb | Se | Zn |
|---|---|---|---|---|---|---|---|---|---|---|
| 500 | 19.4 | 0.69 | 81.3 | 33.7 | 0.13 | 684 | 73.9 | 30.2 | 0.47 | 128 |
| 1000 | 21.1 | 0.61 | 86.6 | 34.6 | 0.11 | 712 | 75.6 | 27.6 | 0.45 | 123 |
| 1500 | 19.9 | 0.61 | 85.0 | 36.0 | 0.12 | 735 | 76.7 | 25.5 | 0.46 | 124 |
| Standard error of the mean | 0.15 | 0.003 | 0.64 | 0.28 | 0.001 | 0.89 | 0.29 | 0.30 | 0.008 | 0.39 |
| *P* value | <0.01 | <0.01 | <0.01 | <0.01 | <0.01 | <0.01 | <0.01 | <0.01 | 0.252 | <0.01 |

**Notes.**
[a]The mean of the metal concentrations in soil samples at three soil depths crossed with four directions.

## RESULTS

### Heavy metal concentrations in soil

The concentrations of 10 heavy metals: As, Cd, Cr, Cu, Hg, Mn, Ni, Pb, Se, and Zn in the soil are summarized in Table 1. As concentrations ranged from 9.8 mg/kg to 56.4 mg/kg, with a mean value of 20.1 mg/kg, slightly higher than the reference value from the Qinghai-Tibet Plateau, suggesting potential As accumulation in the study area. The mean concentrations of Cd and Hg were significantly elevated compared to the reference values for the Qinghai-Tibet Plateau, indicating moderate Cd pollution and elevated Hg levels. Cr and Cu concentrations were also higher than the reference values, reflecting increased levels of these metals in the soil. Mn concentrations ranged from 593 mg/kg to 831 mg/kg, with a mean value of 711 mg/kg, which was close to the reference level. In contrast, the mean Ni concentration was more than double the reference value, suggesting substantial Ni accumulation. Pb concentrations were comparable to the reference, indicating low levels of lead contamination. No data for Se were available for comparison with the Qinghai-Tibet Plateau. Zn concentrations, however, were higher than the reference value, indicating some level of Zn accumulation.

### Heavy metal concentrations in soil at three distances from the boundary

The metal concentrations in the soil, along with the standard error of the mean (SEM) and *p*-values for each metal at three distances (500 m, 1000 m, and 1500 m) from the park boundary, are presented in Table 2. The effect of distance was significant for most metals ($P < 0.01$, except for Se). However, the variation in metal concentrations across the three distances was smaller compared to the variation observed with soil depth and direction.

The concentrations of Cu, Mn, and Ni increased with distance from the park, while Pb concentrations decreased. Other metals did not exhibit a consistent trend relative to distance from the industrial park. These findings indicate that proximity to the industrial park had a measurable impact on soil contamination, particularly for metals such as Mn, Ni, Zn, and Pb.

### Spatial distribution of heavy metals in soil

The heavy metal concentrations at three soil depths (0–10 cm, 10–20 cm, and 20–30 cm) in the eastern, southern, western, and northern directions are presented in Fig. 3. Significant interactions were observed between soil depth and direction ($P < 0.05$). The concentrations of As (Fig. 3A), Cd (Fig. 3B), Hg (Fig. 3E), Pb (Fig. 3G), Se (Fig. 3I), and Zn (Fig. 3J) generally decreased with increasing soil depth ($P < 0.05$), although this trend varied by direction. For Cr (Fig. 3C), Cu (Fig. 3D), Mn (Fig. 3F), and Ni (Fig. 3H), the changes in concentration with soil depth significantly interacted with direction ($P < 0.05$). In the western direction, Cr, Cu, and Mn concentrations increased with soil depth ($P < 0.05$). Similarly, Cu and Ni concentrations increased with depth in the southern direction ($P < 0.05$) but decreased toward the eastern side ($P < 0.05$). In contrast, no significant changes in metal concentrations were observed with depth in the northern direction ($P > 0.05$).

### Potential ecological risk indexes for heavy metals in soil

The potential ecological risk indices for 10 metals (As, Cd, Cr, Cu, Hg, Mn, Ni, Pb, Se, Zn) across four directions (east, south, west, north) and three distances (500 m, 1000 m, 1500 m) are presented in Table 3. Cd consistently exhibited a high ecological risk across all directions and distances, with particularly elevated risk at 500 m and 1000 m. As showed varying risk levels across distances but remained a major concern, especially in the northern and western directions. Zn posed a high ecological risk in the western and northern directions, with relatively stable risk levels across distances. Hg and Se generally presented lower risk compared to Cd and As, though Hg exhibited significant fluctuations, particularly in the southern direction. For most metals, the ecological risk tended to decrease with increasing distance from the source. However, metals such as Cd and As continued to present significant risks even at greater distances.

### Correlations between the soil concentrations of 10 metals

The Pearson correlation coefficients between the soil concentrations of the 10 metals are presented in Table 4. Significant positive correlations ($P < 0.05$) were observed between the following pairs: As and Hg, Mn, Pb, Se, Zn; Cd and Hg, Pb, Se, Zn; Cr and Cu, Ni; Cu and Pb, Zn; Hg and Mn, Se, Zn; Mn and Se; Ni and Se; Pb and Se, Zn; and Se and Zn. Significant negative correlations ($P < 0.05$) were found between As and Cr; Cd and Cr; Cr and Hg; Cu and Hg; and Mn and Pb. Notably, strong positive correlations were found between As and Hg, as well as Zn and Cd. In contrast, certain metals, such as Cr and Hg, or Zn and Ni, exhibited weaker or negative correlations.

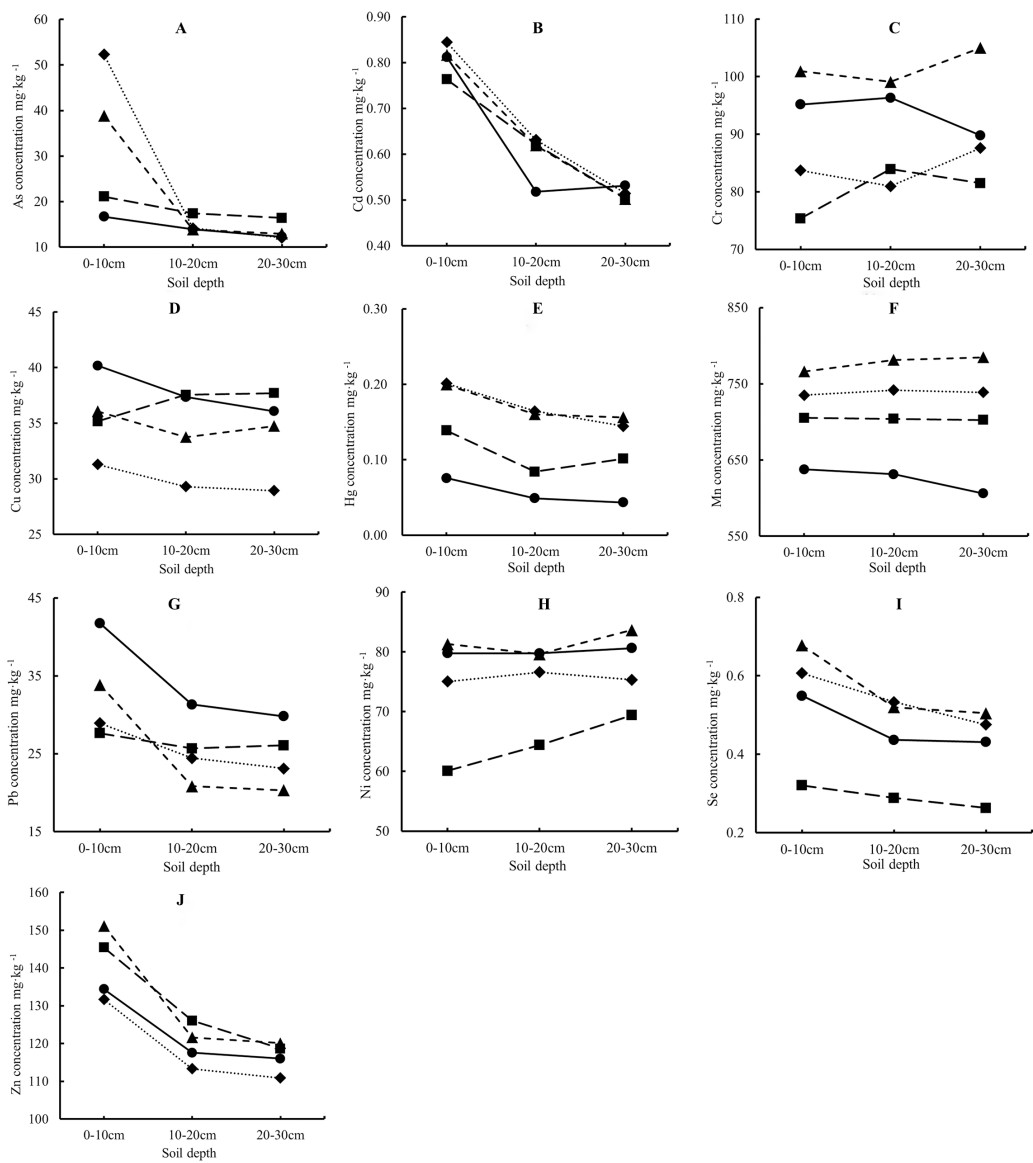

**Figure 3** The concentrations (the means of the three distances from the boundary) of heavy metals at three soil depths (0–10, 10–20, 20–30 cm) to the east, south, west, and north directions of the Park. –●–, East; –■–, South; --▲-- West; ●●◆●●, North. (A) As concentration; (B) Cd concentration; (C) Cr concentration; (D) Cu concentration; (E) Hg concentration; (F) Mn concentration; (G) Pb concentration; (H) Ni concentration; (I) Se concentration; (J) Zn concentration.

## DISCUSSION

The industrial park in this study includes enterprises involved in producing metal-based products such as silicon carbide, chemicals, metallurgical products, and building materials. During the industrial production process, various waste and residual materials are generated and released into the surrounding environment, contributing to potential pollution issues. In this study, we found that the mean concentrations of Cd, Cu, Hg, Ni, Se, and Zn were 1.6

**Table 3 The potential ecological risk indexes of 10 metals.**

| Direction | Distance (m) | Individual ecological risk factor ($E_i$) | | | | | | | | | |
|---|---|---|---|---|---|---|---|---|---|---|---|
| | | As | Cd | Cr | Cu | Hg | Mn | Ni | Pb | Se | Zn |
| | 500 | 10.9 | 83.3 | 5.04 | 2.10 | 5.11 | 6.74 | 4.13 | 6.17 | 62.7 | 0.58 |
| East | 1000 | 12.4 | 81.7 | 4.39 | 2.12 | 5.33 | 6.99 | 3.89 | 6.10 | 71.9 | 0.57 |
| | 1500 | 10.2 | 78.7 | 4.84 | 1.81 | 4.67 | 6.96 | 3.95 | 3.38 | 55.4 | 0.46 |
| | 500 | 14.8 | 91.7 | 3.31 | 1.65 | 14.2 | 7.65 | 2.77 | 3.59 | 37.9 | 0.62 |
| South | 1000 | 13.6 | 68.3 | 3.62 | 1.62 | 5.93 | 7.16 | 2.86 | 3.26 | 36.4 | 0.61 |
| | 1500 | 13.8 | 69.0 | 4.37 | 2.01 | 7.60 | 8.08 | 3.38 | 3.51 | 36.8 | 0.52 |
| | 500 | 24.9 | 86.3 | 5.68 | 1.54 | 13.1 | 7.53 | 4.09 | 3.29 | 81.2 | 0.57 |
| West | 1000 | 30.5 | 85.7 | 4.72 | 2.06 | 14.0 | 8.75 | 4.30 | 5.60 | 73.5 | 0.62 |
| | 1500 | 22.3 | 73.0 | 4.73 | 1.81 | 12.9 | 8.57 | 3.81 | 3.80 | 80.0 | 0.63 |
| | 500 | 33.4 | 84.7 | 4.74 | 1.54 | 12.9 | 7.56 | 3.71 | 3.59 | 71.9 | 0.55 |
| North | 1000 | 37.5 | 82.3 | 3.54 | 1.33 | 12.2 | 7.51 | 3.62 | 3.54 | 66.9 | 0.49 |
| | 1500 | 33.8 | 86.3 | 4.28 | 1.82 | 15.1 | 8.79 | 3.93 | 3.72 | 71.2 | 0.54 |

**Table 4 Correlation coefficients between metal concentrations in soil.**

| | As | Cd | Cr | Cu | Hg | Mn | Ni | Pb | Se |
|---|---|---|---|---|---|---|---|---|---|
| Cd | 0.650[*] | | | | | | | | |
| Cr | −0.274[*] | −0.245[*] | | | | | | | |
| Cu | −0.086 | 0.065 | 0.440[*] | | | | | | |
| Hg | 0.563[*] | 0.472[*] | −0324[*] | −0.378[*] | | | | | |
| Mn | 0.221[*] | 0.097 | 0.056 | −0.067 | 0.728[*] | | | | |
| Ni | −0.024 | −0.123 | 0.464[*] | 0.188 | 0.021 | 0.150 | | | |
| Pb | 0.210[*] | 0.529[*] | 0.020 | 0.562[*] | −0.160 | −0.286[*] | 0.156 | | |
| Se | 0.443[*] | 0.420[*] | 0.036 | −0.155 | 0.508[*] | 0.257[*] | 0.577[*] | 0.236[*] | |
| Zn | 0.460[*] | 0.707[*] | 0.019 | 0.220[*] | 0.310[*] | 0.171 | −0.128 | 0.590 | 0.220[*] |

Notes.
[*]$P < 0.05$.

to 8 times higher than their corresponding reference values for the Qinghai-Tibet Plateau. Se concentrations fell into the Se-rich soil category (0.4 mg/kg) (*Hao et al., 2021*). Notably, the concentrations of Hg (0.04–0.23 mg/kg) and Cd (0.38–0.92 mg/kg) significantly exceeded the reference values of Hg (0.02 mg/kg) and Cd (0.08 mg/kg) for soils in the Qinghai-Tibet Plateau (Table 1). These results indicate that areas near the industrial park are particularly affected by Cd pollution. Other studies have also reported pollution risks from As, Cr, Cu, Hg, Pb, and Zn in soils of the Tibet Plateau (*Hou & Li, 2021*). The main sources of heavy metal pollution in this region are natural processes, transportation, and mining activities (*Yang et al., 2020*).

Elevated concentrations of Cd, Ni, Pb, and Zn have been recorded along highways with heavy traffic, highlighting transportation as a significant source of metal pollution (*Guo et al., 2016*). As and its compounds are particularly concerning due to their toxicity. They can migrate from soil to crops, accumulate in the human body, and pose serious health risks
(*Zhang, Qian & Zhang, 2020*). These findings emphasize the urgent need for improved industrial practices, stricter environmental regulations, and comprehensive monitoring to reduce the ecological and health risks associated with heavy metal pollution in these regions (*Kumar et al., 2019*).

Analyzing the spatial distribution of metal elements in soil is an effective method for identifying contamination sources and high-pollution hotspots (*Xiao et al., 2020*). Changes in the spatial distribution of metal concentrations with soil depth, distance, and direction from the industrial park can provide clues about pollution sources. A pattern of declining metal concentrations with soil depth typically suggests that external sources are the primary contributors to contamination (*Zhang et al., 2022a*; *Zhang et al., 2022b*; *Zhang et al., 2022c*), whereas consistent metal concentrations with depth indicate that the levels are dominated by the natural background content in the soil (*Meng et al., 2021*). In this study, we observed that concentrations of As, Cd, Hg, Pb, Se, and Zn decreased with soil depth in all directions (Table 2), though the extent of this trend varied by direction. This suggests that these metals likely originate from external pollution sources (*Yang et al., 2022*), particularly from manufacturers in the industrial park that process metal-related products. The relatively higher concentrations of As, Cd, Hg, and Se to the north of the park may be attributed to the prevailing southeast winds in the region, which likely transport these elements northward (*Tuyarila et al., 2019*). The declining concentrations of Cd and Hg with increasing distance from the park further support the hypothesis that the industrial park is the primary external source of these metals (*Giarikos et al., 2023*) (Table 2; Fig. 3).

Elemental concentration-correlation analysis is a useful tool for predicting the origin of heavy metals in soil, with significant correlations indicating a potential common source (*Zhao et al., 2022*). Highly significant positive correlations (correlation coefficient > 0.50) were observed for the following pairs: As-Cd, As-Hg, Cd-Pb, Cd-Zn, Cu-Pb, Hg-Mn, Hg-Se, Ni-Se, and Pb-Zn. These correlations suggest that Cd, As, Pb, and Zn may share a common source, as do Hg, As, Mn, and Se, and Pb, Cd, Cu, and Zn (Table 4). The sources of these heavy metals are likely linked to industrial activities, where specific metals are processed and generate by-products or wastes containing heavy metals. These metals enter the soil through atmospheric deposition of airborne pollutants and from solid waste streams (*Bai et al., 2014*).

Soils derived from ultramafic rocks are typically enriched with As, Cr, and Ni (*Zhang et al., 2011*). A similar natural abundance and correlation pattern for these metals is commonly found in the Earth's crust, particularly in Tibetan soils (*Zhang, Deng & Yang, 2002*). Previous studies have shown that the concentrations of Cu, Cr, and Ni in the southwest to northeast regions of the Tibetan Plateau are primarily due to the weathering of bedrock and parent material (*Du et al., 2023*). These findings support our observations, as Cr and Ni concentrations in the soil showed no significant variation with depth or distance from the industrial park.

Cd and Zn concentrations are often linked to both traffic pollution and soil weathering, while Pb is closely associated with traffic pollution (*Li et al., 2024*). This explains the observed decline in Cd and Pb concentrations with increasing soil depth and distance from the park in this study. Additionally, we found that Mn had the highest concentration in the

soil, followed by Zn, with minimal changes in concentration with soil depth or distance. This suggests that the soil's parent material is the primary source of these elements (*Wang et al., 2023*). Our results align with previous reports that Mn is considered a conservative element in soil (*Bai et al., 2014*).

In this study, we found that the potential ecological risk index ($E_r^i$) for Cd ranged from 68.33 to 91.67, and for Se, from 36.38 to 81.15. These values indicate moderate potential ecological risk ($40 \leq E_r^i \leq 80$) at some sites, and in certain locations, the risk was higher, reaching the range of potential ecological risk ($80 \leq E_r^i \leq 160$). The indices for the other eight metals were all below 40, indicating low potential ecological risk (Table 3). Metals in soil can leach into groundwater and surface water, and be absorbed by plants, potentially leading to ecological pollution at high concentrations (*Zhao et al., 2012*).

One key factor affecting the ecological risk of metal elements is the elevated uptake rates of some metals by plants (*Yu & Chen, 2023*). In addition to analyse metal concentrations in soil, we also measured the concentrations of metals in the roots and above-ground parts of the dominant grass species at the sampling sites (*Sai, 2023*). The plant-to-soil concentration ratios were 0.10 for As, 0.90 for Cd, 0.41 for Cu, 0.22 for Hg, 0.14 for Mn, 0.10 for Pb, 0.16 for Se, and 0.62 for Zn. This indicates that Cd had the highest plant uptake rate, followed by Zn and Cu, aligning with the findings of *Hu (2007)*. The reason for the high uptake rate of Cd by plants in this environment remains unclear. However, previous research suggests that Cd is one of the most mobile elements in soil, making it readily absorbed by plants (*Anisimov et al., 2023*). Additionally, Cd has been shown to interact antagonistically with Mn, with Cd accumulation in plants occurring under Mn deficiency (*Rahman et al., 2016*). The similarities in atomic structure and electronegativity between Cd and Zn may contribute to the non-selective uptake of these metals by plants, as Zn is an essential trace element in plants (*Zhao et al., 2010*; *Huang et al., 2023*). Given its high concentration in the soil and high uptake rate by plants, Cd poses a significant risk as a pollutant in this study region.

## CONCLUSION

The soil surrounding the Tianzhu Industrial Park shows elevated levels of Cd, Cu, Hg, Ni, Se, and Zn compared to the reference soil values for the Qinghai-Tibet Plateau. Notably, the Cd concentration exceeds the maximum allowable levels for agricultural soils in China, raising significant concern. The Se content classifies the soil as Se-rich. While Se poses a moderate potential ecological risk, the Cd content presents a considerable ecological threat, particularly due to its high uptake rate by plants. The concentrations of As, Cd, Hg, Pb, Se, and Zn in the soil appear to be primarily influenced by external sources, with variations observed depending on proximity to the industrial park. Manufacturing activities within the park and high-density traffic in the area are likely contributors to the metal pollution.

### Funding
The work was supported by the China Forage and Grass Research System (CARS-34) and the National Natural Science Foundation of China (No. 31660684). The funders had no role in study design, data collection and analysis, decision to publish, or preparation of the manuscript.

### Grant Disclosures
The following grant information was disclosed by the authors:
China Forage and Grass Research System: CARS-34.
National Natural Science Foundation of China: No. 31660684.

### Competing Interests
The authors declare there are no competing interests.

### Author Contributions
- Juan Qi conceived and designed the experiments, performed the experiments, analyzed the data, prepared figures and/or tables, authored or reviewed drafts of the article, and approved the final draft.
- Xin Lu performed the experiments, analyzed the data, prepared figures and/or tables, authored or reviewed drafts of the article, and approved the final draft.
- Ninggang Sai performed the experiments, analyzed the data, prepared figures and/or tables, authored or reviewed drafts of the article, and approved the final draft.
- Yanjun Liu performed the experiments, prepared figures and/or tables, authored or reviewed drafts of the article, and approved the final draft.
- Wangyi Du analyzed the data, prepared figures and/or tables, authored or reviewed drafts of the article, and approved the final draft.

### Data Availability
The raw data is available in the Supplemental Files.

### Supplemental Information
Supplemental information for this article can be found online at http://dx.doi.org/10.7717/peerj.18510#supplemental-information.

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
