# Peer review of "Heavy metal concentrations in soil and ecological risk assessment in the vicinity of Tianzhu Industrial Park, Qinghai-Tibet Plateau"

_PeerJ, doi:10.7717/peerj.18510_

## Round 0.1 · original submission · Major Revisions

The authors are particularly advised to further clearly state the need as well as objectives of this study.

Reviewer 1 ·

Basic reporting

The manuscript present a good data but need further amendments on the language checking, references and need additional elaborations.
Please check the English as there are still some grammatical errors (prepositions, spelling, spacing, commas, small/capital letters, etc.). I would suggest the authors run for proofreads for the English. Please check.
Lines 72-76. The objectives of the studies need to be mentioned in a sentence instead of numbering. Please rephrase and revise.
Lines 72-73. Please mention what is As and Cd too. The palladium (Pd) was mentioned in the objectives, but the results were not provided. The abstract and the study both mention Ni, but the objective does not. Please justify.

Experimental design

Some additional information are needed in the experimental and materials section. Please refer other comments.
Lines 79-82. Why did the authors choose this area or sample sites for the study? Could you provide a specific reason for choosing this area or sampling sites over other nearby areas? Please justify.
Lines 82 and 91. Please be consistent with using either Fig. or Figure in the text. Additionally, the text should align with the actual figure.
Line 96. Any specific reason why did the authors choose the air-dried method and the period of drying to 2 weeks? What happened, or if it is oven-dried?
Please include the materials subsection and state what materials and chemicals have been used in this study.

Validity of the findings

The findings of this study has been assessed and investigated. Further elaboration are needed.

Please elaborate more on the results section based on the data from the tables and figures.
Lines 158, 222-223. Please check that the results stated in the abstract were the same as those mentioned in the text.
Line 170. Please elaborate on why the concentrations were high for these metals in your study.
Please insert the table or figure in the Discussion part so that the authors can understand what the authors are referring to.
Lines 199-200. Please rephrase.
Line 230-231. Please rephrase.

Additional comments

Figure 3: Please insert the label and the unit in the x and y-axis.
Table. 1. Please use NA (not available) or ‘–’ instead of /.
Table 2. I would suggest renaming References 1 and 2 in the table with what they are referring to, as in the notes below the table. Reference 1 to Sheng et al., 2012; Reference 2 to (GB15618-2018; Wang et al. 2017).
Table 3: Please insert the unit of distance.
Please check the style of the references as there are still technical errors (e.g., article titles and journal titles, capital or small letters). Example: Line 390, and Ref Zhang XR et al., 2022. Please follow the journal’s guidelines. Also, please try to reduce the references. The authors can omit outdated references and include references from the PeerJ journals.

Reviewer 2 ·

Basic reporting

In this study, the concentrations of 10 metal elements were analyzed in soils around the Tianzhu Industrial Park on the Tibetan Plateau. They found that the concentrations of arsenic, cadmium, mercury, lead, selenium and zinc varied with soil depth and orientation, suggesting an association with specific producers in the park. Until now, there have been relatively few studies of soil contamination around soil industrial parks on the Tibetan Plateau. However, the English writing of this paper needs further polishing. Therefore, this article is recommended for acceptance with major revisions. The following are some of my recommendations:

Suggestions:
Further polish the language throughout the text.
Line 29: What are the units in these sentences, mg/kg or multiples of higher.
Line 34: That sentence doesn't quite fit at the end seems out of place.
Line 105: The standard limits in this section refer to China's agricultural soil standards. However, it does not look like agricultural soil in Figure 2.
Line 136: Reference 2 are the metal concentrations for agricultural soils in China (GB15618 -2018; Wang et al. 2017). The standard was not cited correctly. The limits of this standard are closely related to soil pH, are there any soil related physic-chemical property data available for this study?

Experimental design

The standard limits in this section refer to China's agricultural soil standards. However, it does not look like agricultural soil in Figure 2.

Validity of the findings

/

Additional comments

/

---

## Round 0.2 · Minor Revisions

As you can see both the reviewers are satisfied with the revised manuscript. However, a couple of minor points need to be clarified. I urge you to do it immediately to further process this article.

Reviewer 1 ·

Basic reporting

1.Please include a reason on why you choose the site in the text.
2.Please mention what is the SD stand for in Table 1 and SEM in Table 2.

Experimental design

OK

Validity of the findings

OK

Reviewer 2 ·

Basic reporting

The author's response addressed my concerns and I had no further questions.

Experimental design

/

Validity of the findings

/

Additional comments

/

---

## Round 0.3 · accepted · Accept

Accept my felicitations for conducting this nice study.